 **eLIFE**

# C. elegans avoids toxin-producing Streptomyces using a seven transmembrane domain chemosensory receptor

Alan Tran[1], Angelina Tang[1], Colleen T O'Loughlin[2], Anthony Balistreri[3†], Eric Chang[1†], Doris Coto Villa[1†], Joy Li[1†], Aruna Varshney[1†], Vanessa Jimenez[1], Jacqueline Pyle[1], Bryan Tsujimoto[1], Christopher Wellbrook[1], Christopher Vargas[1], Alex Duong[1], Nebat Ali[1], Sarah Y Matthews[3], Samantha Levinson[3], Sarah Woldemariam[4], Sami Khuri[5], Martina Bremer[6], Daryl K Eggers[3], Noelle L'Etoile[4], Laura C Miller Conrad[3]*, Miri K VanHoven[1]*

[1]Department of Biological Sciences, San Jose State University, California, United States; [2]Department of Bioengineering and Therapeutic Sciences, University of California San Francisco, San Francisco, United States; [3]Department of Chemistry, San Jose State University, California, United States; [4]Department of Cell & Tissue Biology, University of California San Francisco, San Francisco, United States; [5]Department of Computer Science, San Jose State University, California, United States; [6]Department of Mathematics and Statistics, San Jose State University, California, United States

**\*For correspondence:**
laura.miller.conrad@sjsu.edu (LCMC);
miri.vanhoven@sjsu.edu (MKVH)

†These authors contributed equally to this work

**Competing interests:** The authors declare that no competing interests exist.

**Abstract** Predators and prey co-evolve, each maximizing their own fitness, but the effects of predator–prey interactions on cellular and molecular machinery are poorly understood. Here, we study this process using the predator *Caenorhabditis elegans* and the bacterial prey *Streptomyces*, which have evolved a powerful defense: the production of nematicides. We demonstrate that upon exposure to *Streptomyces* at their head or tail, nematodes display an escape response that is mediated by bacterially produced cues. Avoidance requires a predicted G-protein-coupled receptor, SRB-6, which is expressed in five types of amphid and phasmid chemosensory neurons. We establish that species of *Streptomyces* secrete dodecanoic acid, which is sensed by SRB-6. This behavioral adaptation represents an important strategy for the nematode, which utilizes specialized sensory organs and a chemoreceptor that is tuned to recognize the bacteria. These findings provide a window into the molecules and organs used in the coevolutionary arms race between predator and potential prey.

DOI: https://doi.org/10.7554/eLife.23770.001

## Introduction

The evolutionary interactions of predators with their prey have profoundly influenced the history of life on earth (*Lotka, 1920*). To better understand these interactions and their influences at the molecular and cellular levels, we studied the bacteriovore *Caenorhabditis elegans* and several species of the Gram-positive bacterium *Streptomyces* that have evolved strong defenses to avoid predation by nematodes. *Streptomyces avermitilis, S. milbemycinicus,* and *S. costaricanus* all produce potent nematicides or antinematodal compounds (*Miller et al., 1979*; *Burg et al., 1979*; *Egerton et al., 1979*; *Takiguchi et al., 1980*; *Dicklow et al., 1993*; *Esnard et al., 1995*)

(*Figure 1A*). Interestingly, these bacteria have been used to produce drugs against nematode infections, such as avermectin and milbemycin, which paralyze nematodes by irreversibly opening invertebrate glutamate-gated chloride channels (*Cully et al., 1994*; *Holden-Dye and Walker, 2014*). Despite this usage, little is known about whether or how *C. elegans* recognizes and responds to *Streptomyces* in the environment.

## Results

To determine whether *C. elegans* responds to *Streptomyces*, we used dry-drop avoidance assays that test the response of either the phasmid sensory neurons in the tail or the amphid sensory neurons in the head (*Park et al., 2011*; *Hilliard et al., 2002*) to *Streptomyces* (*Figure 1B,D*). First, we tested the response to exposure at the tail. Interestingly, animals rapidly halted backwards movement when exposed to *Streptomyces* strains but not *Escherichia coli*, a Gram-negative species used to feed worms in the laboratory, or *Bacillus subtilis*, a Gram-positive species found in soil and vegetation (*WilpatWipat and Harwood, 1999*) (*Figure 1C*). These data indicate that *C. elegans* have the ability to recognize and specifically avoid *Streptomyces.* Few compounds have been fully characterized with respect to these neurons, but dilute concentrations of sodium dodecyl sulfate (SDS) have been shown to elicit a rapid and robust avoidance response in *C. elegans* hermaphrodites (*Hilliard et al., 2002*), the primary sex found in nature (*Brenner, 1974*). The avoidance response to *Streptomyces* is comparable to the response to SDS (*Figure 1C*). We hypothesize that the tail-mediated response to *Streptomyces* is due to a secretion product, as animals responded similarly to cell-free supernatants (*Figure 1C*).

We wondered whether *C. elegans* would demonstrate a similar avoidance response upon encountering *Streptomyces* at their head. Thus, we tested the response to exposure at the head. Animals exposed to a control buffer, *E. coli* or *B. subtilis* at the head usually continued moving forward, but animals exposed to *S. avermitilis*, *S. milbemycinicus* or *S. costaricanus* usually halted movement, avoiding *Streptomyces* (*Figure 1D,E*). These data indicate that *C. elegans* also recognize *Streptomyces* at the head. *C elegans* did not demonstrate avoidance in chemotaxis assays designed to test responses to volatile odorants, suggesting that direct contact is required (*Figure 1—figure supplement 1*).

SDS is a laboratory detergent, but the very similar responses of *C. elegans* to *Streptomyces* and SDS led us to hypothesize that *Streptomyces* might secrete a small molecule similar to SDS in activity or structure. The avoidance could be due to a non-specific interaction that is based on the compound's surfactant properties, or it could be due to specific binding and recognition of SDS. We found that the surfactant activity of SDS was not required for the avoidance response, as concentrations ten-fold below the critical micelle concentration (*Rahman and Brown, 1983*) were still avoided (*Figure 2A*, *Figure 2—figure supplement 1*, *Figure 2—figure supplement 2*). Similarly, a surfactant (Triton X-100) at its critical micelle concentration (*Tiller et al., 1984*) did not elicit a strong avoidance response (*Figure 2A*). These findings indicate that SDS is unlikely to elicit a response through a non-specific detergent effect. Therefore, we looked to define the structural elements of SDS that are responsible for the avoidance response.

To determine which structural elements of SDS are required for avoidance, we tested whether the hydrocarbon chain or the polar head group in SDS was required to induce avoidance. Sodium hexyl sulfate, which has a carbon chain half the length of SDS, elicited reduced chemosensation (*Figure 2B*). 1-Dodecanol, which has a smaller, less polar head group but the same carbon chain length, was not sensed by the phasmid neurons (*Figure 2B*). These findings indicate that both the tail length and the nature of the polar head group are important for the response.

Although SDS does not occur in nature, carboxylic acids with similar amphipathic structures do. Both SDS and these carboxylic acids have long hydrophobic alkyl tails and negatively charged head groups at pH 7. Interestingly, the NP10 strain of *Streptomyces* (*Figure 1A*) secretes carboxylic acids that have these characteristics (*Ilic-Tomic et al., 2015*). We therefore used the tail drop assay to test a series of carboxylic acids of varying chain lengths, with and without *cis* double-bonds (*Figure 2C* and *Figure 2—figure supplement 1*), to determine which, if any, *C. elegans* avoid. Dodecanoic acid and decanoic acid induced a robust avoidance response in the tail (*Figure 2C*, *Figure 2—figure supplement 2*). Additionally, these two compounds induce a robust avoidance response in the head

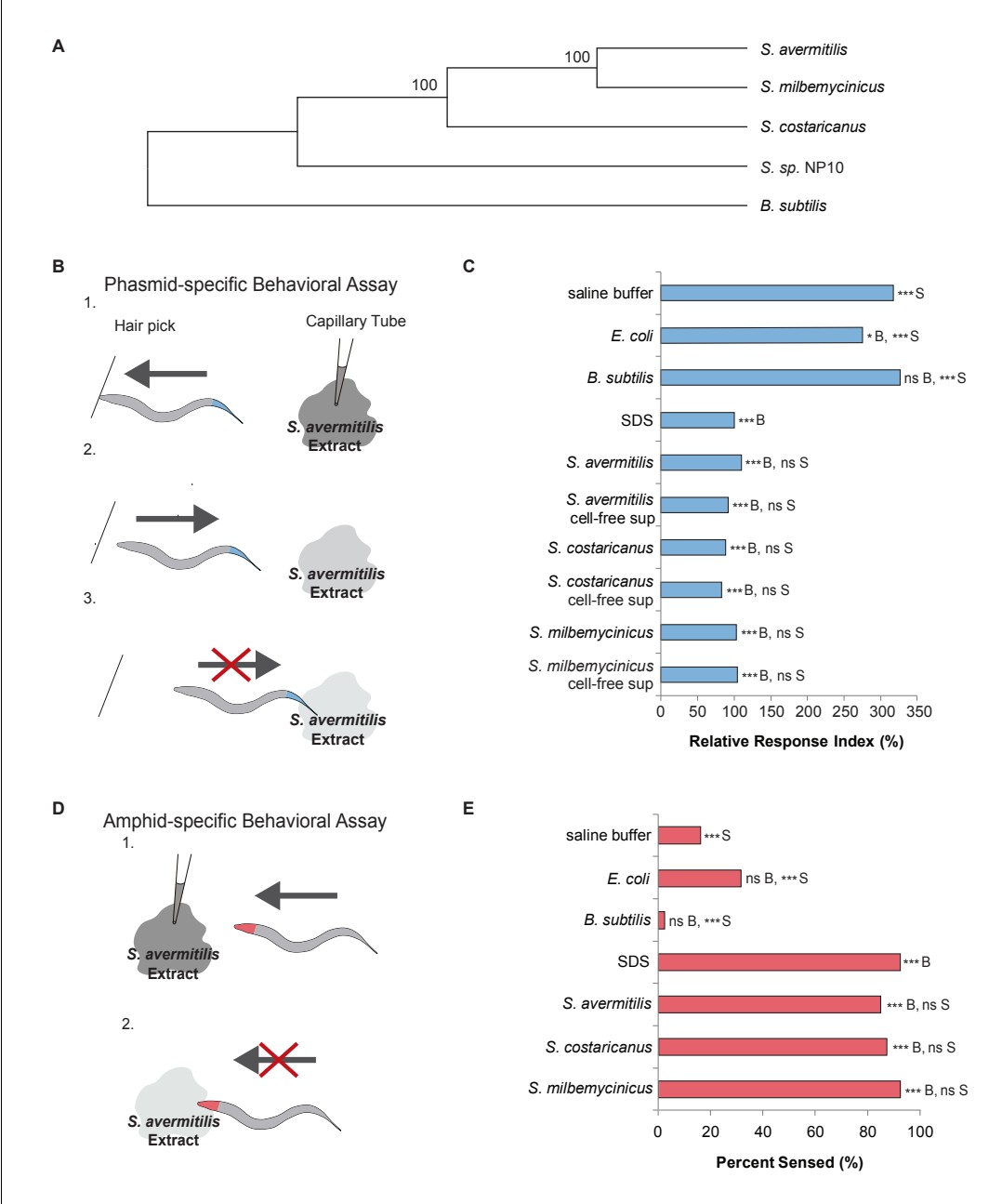

**Figure 1.** *C. elegans* nematodes rapidly avoid *Streptomyces* at their head and tail. (**A**) Phylogenetic tree including strains of *Streptomyces* bacteria and *Bacillus subtilis*. (**B**) Schematic diagram of a dry-drop behavioral assay that selectively tests the response of the PHA and PHB phasmid neurons. Animals are placed on a dry plate and induced to move backward with a hair pick. A drop of bacterial extract or control buffer is placed behind the animal and quickly dries, preventing wicking along the animal. As the animal moves into the drop, the time before the animal halts movement is measured. Response times are normalized to a positive control, the average response to sodium dodecyl sulfate (SDS), assayed on the same day. A shorter response time indicates a more rapid avoidance response. (**C**) Quantification of the rapid avoidance response of *C. elegans* phasmid neurons to *Streptomyces*, but not to *Escherichia coli* or *B. subtilis*. One-way ANOVA and Tukey's post-hoc tests were performed to determine significance. (**D**) Schematic diagram of a dry-drop behavioral assay that selectively tests the response of amphid neurons in the head. Animals are placed on a dry plate and allowed to move forward. A drop of bacterial extract or control buffer is placed in front of the animals and quickly dries. The proportion of the time that animals halt movement into the dry drop is recorded. (**E**) Quantification of the rapid avoidance response of *C. elegans* amphid neurons to species of *Streptomyces*, but not *E. coli* or *B. subtilis*. Two independent sample z-tests and the Hochberg multiple comparison adjustment procedure were performed. (**C and E**) For each experimental sample, n ≥ 40 for the experimental group, n ≥ 40 for the 0.6% SDS positive control, and n ≥ 40 for the saline buffer negative control. ***B, $p<0.001$; *B, $p<0.05$; ns B, not significant compared with a control buffer; ***S, $p<0.001$; NS S, not

*Figure 1 continued on next page*

*Figure 1 continued*

significant compared with SDS positive control. Exact values and additional pairwise comparisons are included in *Figure 1—source data 1* and *Figure 1—source data 2*.

DOI: https://doi.org/10.7554/eLife.23770.002

The following source data and figure supplement are available for figure 1:

**Source data 1.** Source data for *Figure 1* and *Figure 1—figure supplement 1*.

DOI: https://doi.org/10.7554/eLife.23770.004

**Source data 2.** Significance for pairwise comparisons for *Figure 1*.

DOI: https://doi.org/10.7554/eLife.23770.005

**Figure supplement 1.** *C. elegans* do not avoid the smell of *Streptomyces.*

DOI: https://doi.org/10.7554/eLife.23770.003

(*Figure 2D*). Thus, sensation of these molecules in particular could contribute to the recognition and avoidance of *Streptomyces* bacteria by *C. elegans*.

To determine whether strains of *Streptomyces* that secrete toxins also secrete dodecanoic acid or decanoic acid, we extracted and labeled the fatty acids secreted from *Streptomyces* strains and *B. subtilis*. We used high-resolution mass spectrometry to determine relative production levels in cell-free supernatants (*Figure 3*, *Figure 3—figure supplement 1*). We found that *S. avermitilis, S. costaricanus* and *S. milbemycinicus* did indeed secrete dodecanoic acid, whereas dodecanoic acid secretion from *B. subtilis* was not detected above background. A lower signal for decanoic acid was also detected. Although additional cues from *Streptomyces* probably contribute to the *C. elegans* response, our observation of dodecanoic acid supports the hypothesis that this molecule is a cue that is produced by *Streptomyces*.

Given that the cue from *Streptomyces* is secreted, we hypothesized that a chemosensory receptor might mediate the avoidance response. There are two pairs of chemosensory neurons in the *C. elegans* tail that have exposed dendrites: the PHA and PHB chemosensory neurons (*White et al., 1986*; *Hedgecock et al., 1985*). These neurons mediate the rapid avoidance of SDS and other chemicals (*Hilliard et al., 2002*). In the head, additional neurons play roles in chemical avoidance, including the polymodal nociceptor ASH (*Bargmann, 2006*). Therefore, to identify the cognate receptor for dodecanoic acid, we performed an expression screen using existing data that are available on Wormbase.org (*WormBase web site, 2017*) for chemoreceptor genes that are expressed in the PHA and PHB neurons and in at least one set of head or amphid neurons. We performed RNA interference (RNAi) (*Fraser et al., 2000*; *Kamath and Ahringer, 2003*) to knock down the function of these genes and screened for defects in the phasmid response to dodecanoic acid. We ranked the chemoreceptor genes by the magnitude of the defect, and focused our work on the gene whose knock-down resulted in the most severe phenotype, that encoding the SRB-6 G protein-coupled receptor (GPCR) (*Figure 4—figure supplement 1*). *srb-6* knock-down also resulted in the most severe defect among knockdowns of *srb* family members (*Figure 4—figure supplement 1*). In addition to *srb-6*'s expression in PHA and PHB neurons in the tail, this gene is expressed in only three sets of sensory neurons in the head; *srb-6* is highly expressed in the ASH and ADL neurons, and weakly expressed in the ADF neurons (*Troemel et al., 1995*).

To further test the role of *srb-6* in mediating the rapid avoidance response, we performed behavioral assays with *srb-6* presumptive null mutants (*Thompson et al., 2013*). *srb-6* mutants were defective in the tail (*Figure 4A*) and head (*Figure 4B*) avoidance response to species of *Streptomyces* and to dodecanoic acid. These data are consistent with the requirement of this GPCR in the avoidance of both *Streptomyces* and the fatty acid secreted by these bacteria. In the tail, the limited expression of the *srb-6* GPCR to PHA and PHB neurons indicates that *srb-6* might function in these neurons. To test this, we cloned the *srb-6* cDNA and expressed it under the direction of the *ocr-2* promoter. Within the tail region, this promoter selectively drives expression in the PHA and PHB neurons (*Tobin et al., 2002*). Expression of *srb-6* in PHA and PHB significantly restored the *srb-6* mutants' ability to sense both *S. avermitilis* and dodecanoic acid at the tail (*Figure 4C*). Taken together, the endogenous expression of *srb-6* in PHA and PHB neurons, the ability of *srb-6* expression in PHA and PHB to rescue *srb-6* behavioral defects, and the role of PHA and PHB as the sole chemosensory neurons in the tail, indicate that the *srb-6* GPCR probably functions in one or both of these cells.

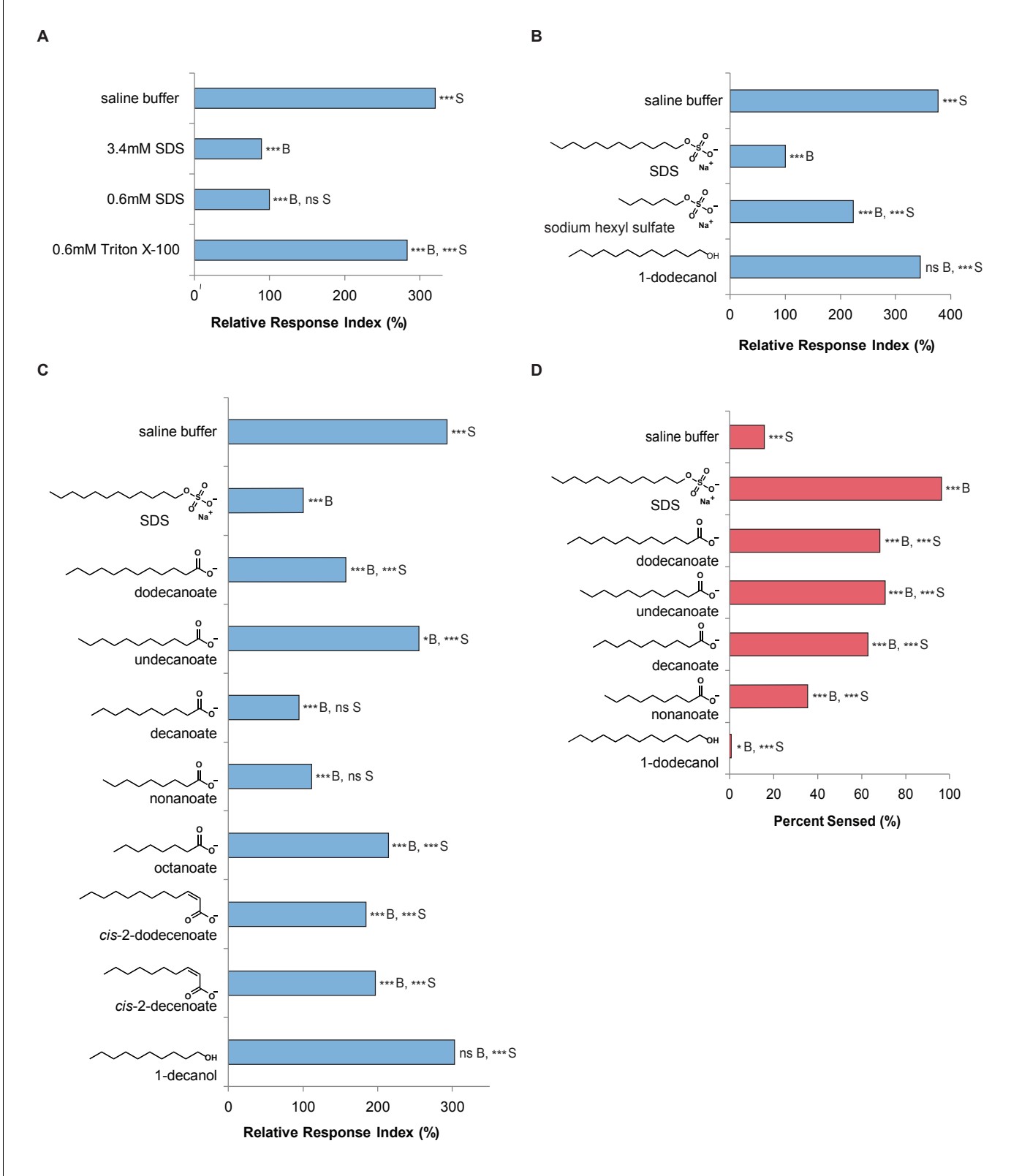

**Figure 2.** *C. elegans* avoids the SDS analogs dodecanoate and decanoate. (**A**) Quantification of the similar phasmid responses to 3.4 mM and 0.6 mM SDS, which is ten-fold below the critical micelle concentration of SDS. All other assays in this study were performed with 0.6 mM SDS to preclude the possibility of surfactant activity. The surfactant Triton X-100 at its critical micelle concentration is not strongly sensed by the phasmid neurons. (**B**) Quantification of the reduced responses to molecules with shorter tail length or a less polar head group than SDS. SDS and sodium hexyl sulfate were

*Figure 2 continued on next page*

*Figure 2 continued*

tested at 0.6 mM and 1-dodecanol was tested at 1 mM. (**C**) Quantification of the similar phasmid responses to SDS, dodecanoate, decanoate and nonanoate. All compounds except SDS were tested at 1 mM. Note that compounds are represented as the predominant form in a pH 7 solution, the carboxylates. (**D**) Quantification of the similar amphid responses to SDS, dodecanoate, decanoate and undecanoate, and slower responses to nonanoate and 1-decanol. All compounds except SDS were tested at 1 mM. Note that carboxylate forms are again represented. (**A–C**) For phasmid assays, one-way ANOVA and Tukey's post-hoc test were performed. (**D**) For amphid assays, two-sample z-tests and the Hochberg multiple comparison adjustment procedure were performed. (**A–D**) For all experimental samples, $n \geq 40$ for the experimental group, $n \geq 40$ for the 0.6% SDS positive control, and $n \geq 40$ for the saline buffer negative control. ***B, $p<0.001$; *B, $p<0.05$; ns B, not significant compared with a control buffer; ***S, $p<0.001$; ns S, not significant compared with SDS positive control. Exact values and additional pairwise comparisons are included in *Figure 2—source data 1* and *Figure 2—source data 2*.

DOI: https://doi.org/10.7554/eLife.23770.006

The following source data and figure supplements are available for figure 2:

**Source data 1.** Source data for *Figure 2* and *Figure 2—figure supplement 2*.
DOI: https://doi.org/10.7554/eLife.23770.009
**Source data 2.** Significance for pairwise comparisons for *Figure 2* and *Figure 2—figure supplement 2*.
DOI: https://doi.org/10.7554/eLife.23770.010
**Figure supplement 1.** CAS numbers for compounds purchased for this study.
DOI: https://doi.org/10.7554/eLife.23770.007
**Figure supplement 2.** Response to different concentrations of SDS, dodecanoate and decanoate.
DOI: https://doi.org/10.7554/eLife.23770.008

In the head, the *ocr-2* promoter drives expression in ASH, ADL and ADF gustatory neurons in addition to AWA olfactory neurons (*Tobin et al., 2002*). Using this promoter, we found that expression in this subset of neurons significantly rescued the ability of *srb-6* mutants to respond to *S. avermitilis* and dodecanoic acid exposure at the head (*Figure 4D*). Although we cannot rule out a function in AWA, the endogenous expression of *srb-6* in ASH, ADL and ADF (and not AWA) suggests a role in one or more of these cells.

To further test whether *srb-6* could act as a receptor for both dodecanoic acid and cell-free supernatants from *S. avermitilis*, we visualized calcium changes in ASH neurons in wild-type and *srb-6* mutant animals (*Figure 4E,F*). We found that calcium influx in response to both dodecanoic acid and *S. avermitilis* was severely reduced in *srb-6* mutants, consistent with this GPCR having the ability to detect these stimuli.

The GPCR required for this avoidance response provides *C. elegans* with the ability to avoid potentially deadly prey. In order to understand the broader evolutionary context for this gene, we performed phylogenetic analysis of the *srb-6* receptor. The *srb* family of chemoreceptors is nematode-specific (*Troemel et al., 1995*; *Chen et al., 2005*), consistent with a role for *srb-6* in mediating nematode avoidance of *Streptomyces*. Phylogenetic analysis of the *srb-6* receptor indicates that its closest homologs are within the *Caenhorabditis* genus. Although the sequences for most nematodes are not complete, the next monophyletic group contains several other species of nematodes with life cycles that include free-living soil-dwelling stages (*Figure 4—figure supplement 2*), suggesting possible conservation of this mechanism.

## Discussion

Our results demonstrate that the bacteriovore *C. elegans* detects and escapes from *Streptomyces* species that secrete powerful toxins. *C. elegans* rapidly avoids *Streptomyces* secretions, which include dodecanoic acid. Interestingly, dodecanoic acid itself has been shown to have nematicidal activity (*Tarjan and Cheo, 1956*; *Abdel-Rahman et al., 2008*; *Gu et al., 2005*; *Dong et al., 2014*; *Zhang et al., 2012*). The SRB-6 GPCR is essential for the rapid avoidance of *Streptomyces* and dodecanoic acid. SRB-6 is expressed in the only two phasmid chemosensory neurons, PHA and PHB, as well as in a subset of chemosensory neurons in the amphids ASH, ADL and ADF (*Troemel et al., 1995*). Rescue of gene expression in these cells and in one additional amphid olfactory neuron largely restores the response of *srb-6* mutants to *Streptomyces* and dodecanoic acid exposure at the tail or head, consistent with a role for *srb-6* in those, or a subset of those, cells. PHA and PHB are the only two chemosensory neurons in the hermaphrodite tail, and they mediate a similar

**Figure 3.** Detection of carboxylates secreted by *S.avermitilis*, *S. costaricanus*, and *S. milbemycinicus*. Carboxylates were extracted from species of *Streptomyces* and labeled with a pyridinium moiety for easier detection by mass spectrometry. For example, to detect dodecanoate after *Streptomyces* cell-free supernatants were extracted and concentrated, dodecanoate and the other carboxylates in the extract were converted to pyridinium esters by treatment with carbonyldiimidazole (CDI) in acetonitrile (ACN).
DOI: https://doi.org/10.7554/eLife.23770.011
The following figure supplement is available for figure 3:

**Figure supplement 1.** Dodecanoic acid is secreted by *Streptomyces*.
DOI: https://doi.org/10.7554/eLife.23770.012

response to dilute SDS. ASH and ADL also mediate similar chemical avoidance responses, and ADF has been shown to play a role in salt chemotaxis (*Hilliard et al., 2002*; *Bargmann, 2006*; *Jang et al., 2012*). Our calcium-imaging experiments directly support a role for ASH in the SRB-6-mediated avoidance response.

The detection and escape from *Streptomyces* bacteria and their secretion, dodecanoic acid, described here are the first environmentally relevant chemosensory functions identified for the hermaphrodite phasmid circuit that inform our understanding of interactions with bacteria. Interestingly, a critical function of PHB phasmid sensory neurons in males has recently been discovered: sensing hermaphrodite-derived mating cues (*Oren-Suissa et al., 2016*). Synapses between PHBs and their normal partners in hermaphrodites are eliminated early in development, and PHBs form male-specific circuits that regulates this distinct function (*Oren-Suissa et al., 2016*). The hermaphrodite phasmid circuit regulates all chemosensation in the tail, suggesting that these sensilla may have evolved to allow the nematodes to avoid *Streptomyces* and other environmental threats. Interestingly, dodecanoic acid is secreted by crown daisy plants (*Chrysanthemum coronarium L.*), which are intercropped with tomatoes (*Solanum lycopersicum*) to prevent parasitism by root-knot nematodes (*Meloidogyne incognita*), and root-knot nematodes are repelled by 4 mM dodecanoic acid (but not by lower concentrations) (*Dong et al., 2014*). This, in addition to our phylogenetic analyses, suggests conservation among nematodes of this mechanism of escaping toxin-producing organisms that would otherwise be a good food source.

In addition to the detection of dodecanoic acid, we also discovered that decanoic acid elicits a strong escape response in *C. elegans* that is also mediated by the SRB-6 GPCR. However, high levels of decanoic acid are not secreted by *Streptomyces*. Decanoic acid has also been shown to have nematicidal properties (*Tarjan and Cheo, 1956*; *Abdel-Rahman et al., 2008*; *Zhang et al., 2012*), and decanoic acid, dodecanoic acid, and fatty acids of similar lengths are present in many plant

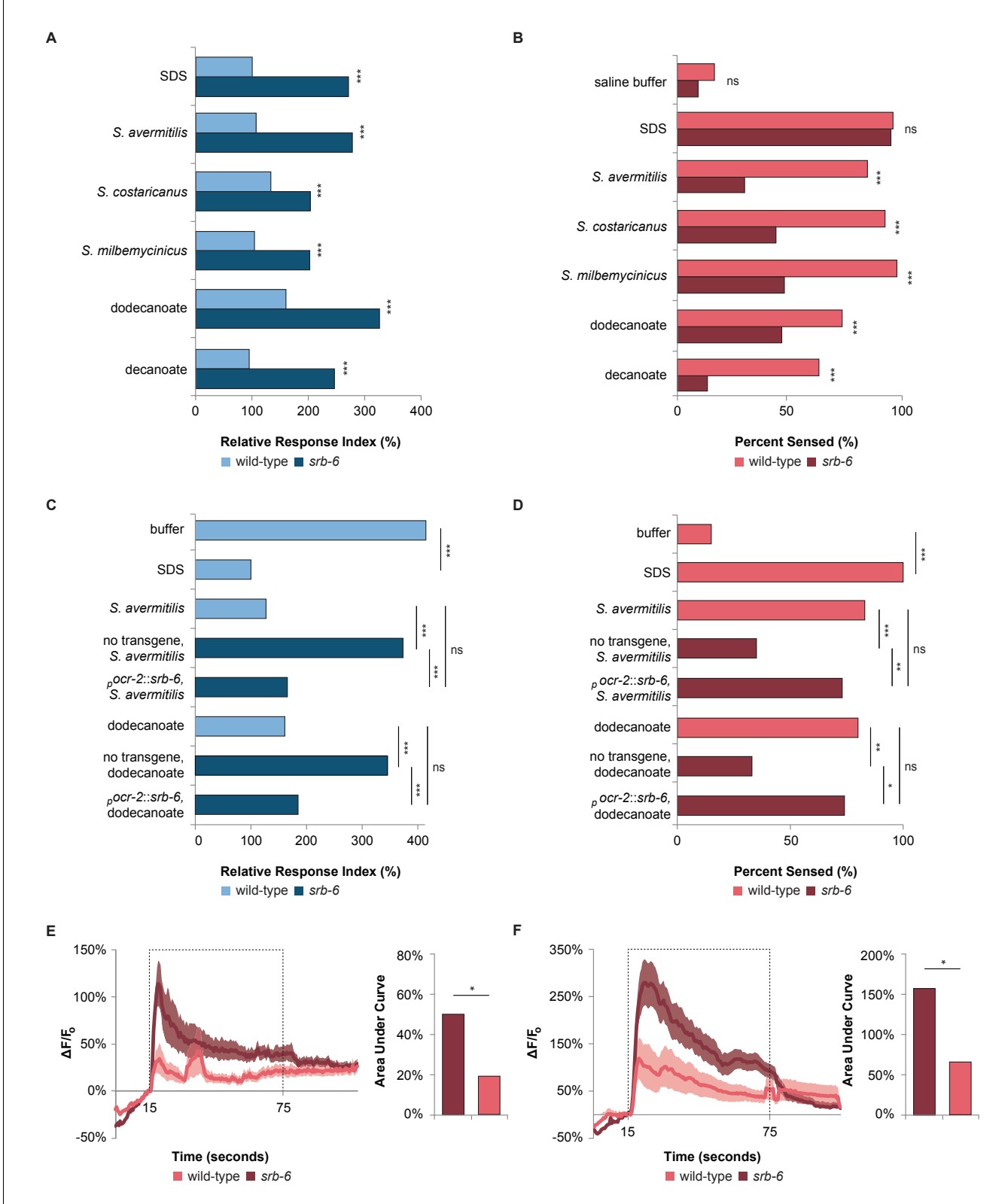

**Figure 4.** The GPCR SRB-6 is required for the response to *Streptomyces*, dodecanoate and decanoate. (**A**) Quantification of the defect in the phasmid response to SDS, *Streptomyces*, dodecanoate and decanoate in *srb-6* null mutants. (**B**) Quantification of the defect in the amphid response to *Streptomyces*, dodecanoate and decanoate in *srb-6* mutants. (**A–D**) For each *Streptomyces* species in each panel, cell-free supernatants were used for wild-type and *srb-6* mutants. (**A–B**) Decanoate and dodecanoate were tested at 1 mM. Note that carboxylate forms are represented. For experimental

*Figure 4 continued on next page*

*Figure 4 continued*

samples, n $\geq$ 40 for the experimental group, n $\geq$ 40 for the 0.6% SDS positive control, and n $\geq$ 40 for the saline buffer negative control. \*\*\*, p<0.001; NS, not significant. (C) Quantification of the significantly rescued response of *srb-6* mutant animals expressing the *srb-6* cDNA under the direction of the *ocr-2* promoter (p*ocr-2*::*srb-6*) to *S. avermitilis* and dodecanoate compared with the response of *srb-6* mutant animals without the transgene. In the tail, the *ocr-2* promoter drives *srb-6* cDNA expression in PHA and PHB neurons. (D) Quantification of the significantly rescued response of *srb-6* mutant animals carrying the p*ocr-2*::*srb-6* transgene to *S. avermitilis* and dodecanoate compared with that of *srb-6* mutant animals without the transgene. In the head, the *ocr-2* promoter drives expression in ASH, ADL, ADF and one other type of amphid neuron (AWA). (C–D) Decanoate and dodecanoate were tested at 1 mM. Note that carboxylate forms are represented. Two similar transgenic lines were examined. For experimental samples, n $\geq$ 20 for the experimental group, n $\geq$ 20 for the 0.6% SDS positive control, and n $\geq$ 20 for the saline buffer negative control. \*\*\*, p<0.001; \*\*, p<0.01; \*, p<0.05; ns, not significant. (A, C) For phasmid assays, one-factor ANOVA analysis was performed followed by two-sample *t*-tests and the Hochberg procedure for multiple comparisons. (B, D) For amphid assays, two-sample z-tests and the Hochberg multiple comparison adjustment procedure were performed. Exact values are included in *Figure 4—source data 1*. (E) Calcium imaging of ASH sensory neurons in living wild-type and *srb-6* mutant animals expressing *GCaMP6* (*Chen et al., 2013*) under the direction of the *srb-6* promoter upon exposure to a cell-free supernatant of *S. avermitilis*. The bar graph shows the average areas under the curve (AUC) for the period of *S. avermitilis* stimulation and error bars are standard error of the mean (SEM). (F) Calcium imaging of ASH neurons in living wild-type and *srb-6* mutant animals expressing *GCaMP6* under the direction of the *srb-6* promoter upon exposure to 1 mm dodecanoate. The bar graph represents the average AUC for the period of dodecanoate stimulation, and error bars are SEM. (E–F) Experimental animals were exposed to a control buffer for the first 15 s, then to stimulus for 60 s, followed by control buffer. Traces show the average percent change in *GCaMP6* fluorescence ($\Delta F/F_0$) and shading indicates SEM. For experimental samples, n $\geq$ 15 for each genotype and stimulus. \*, p<0.05 in a t-test. See Statistical Analysis in the Materials and methods section for additional details.

DOI: https://doi.org/10.7554/eLife.23770.013

The following source data and figure supplements are available for figure 4:

**Source data 1.** Source data for *Figure 4* and *Figure 4—figure supplement 1*.
DOI: https://doi.org/10.7554/eLife.23770.016
**Source data 2.** Significance for pairwise comparisons for *Figure 4—figure supplement 1*.
DOI: https://doi.org/10.7554/eLife.23770.017
**Figure supplement 1.** RNAi screen for receptors required to sense dodecanoic acid.
DOI: https://doi.org/10.7554/eLife.23770.014
**Figure supplement 2.** *C. elegans srb-6* has homologs within the *Caenorhabditis* genus and the nematode phylum.
DOI: https://doi.org/10.7554/eLife.23770.015

seeds (*Zhang et al., 2012*), indicating that avoiding decanoic acid in the environment may also be important to nematodes, outside the context of bacterial avoidance.

The more well-studied interaction between *C. elegans* and bacterial species is the host–pathogen interaction, in which bacteria infect *C. elegans*. *Caenorhabditis elegans* is frequently attracted to pathogens initially, facilitating colonization of the host (*Schulenburg et al., 2007*; *Meisel and Kim, 2014*). In the predator–prey interaction disclosed in the present work, however, we demonstrate that *C. elegans* has evolved a mechanism to recognize and escape from the powerful defenses presented by *Streptomyces*. This avoidance mechanism probably arose through selection pressure on GPCRs, and by employment of reflex-like neural circuits in the head and tail sensory organs, a mechanism that may be common to other predator–prey interactions. Thus, this study deepens our understanding of the molecules, neural circuitry, and behavior used in the arms race between predators and their prey.

## Materials and methods

### *Caenorhabditis elegans* strains

Wild-type C. *elegans* strains were the Bristol strain N2 or *wyIs157 IV ox9* (*10*). Strains were maintained using standard methods (*Brenner, 1974*). Worms were grown at 20°C and on NGM plates seeded with OP50 *E. coli* (RRID:WB-STRAIN:OP50). Mutations used in this study include *eri-1* (*mg366*) (*Kennedy et al., 2004*), and *srb-6*(*gk925263*) *II* (*24*), which was outcrossed four times by pushing out the linked marker *unc-4*(*e120*) *II; wyIs157 IV ox4*(*10*). *srb-6*(*gk925263*) *II* is likely to be null, as it has an early stop codon in *srb-6*: Y186Amber. The outcrossed *srb-6*(*gk925263*) *II; wyIs157 IV ox4* strain was sequenced to verify the substitution mutation. Transgenes used for rescue of *srb-6* and maintained as extrachromosomal arrays are *iyEx348* and *iyEx350* (30–50 ng/ul p*ocr-2*::*srb-6*, 40 ng/ul p*odr-1*::GFP [*L'Etoile and Bargmann, 2000*]), and 'no transgene' *srb-6* animals were *odr-1*::

*GFP* negative animals from *iyEx348*; *srb-6* and *iyEx350*; *srb-6*. *ocr-2* is expressed in PHA and PHB neurons in the tail, and in ASH, ADL, ADF and AWA neurons in the head (*Tobin et al., 2002*). The transgene used for calcium imaging is *iyEx358* (30 ng/ul $_p$*srb-6*::GCaMP6 and 20 ng/ul $_p$*unc-22*:: *DsRed*).

### Bacterial strains and media

For solid cultures, *Streptomyces* was grown on mannitol-soy agar at 28°C. *Bacillus subtilis* was grown on mannitol-soy agar or trypticase soy agar (TSA) at 37°C. *Escherichia coli* for drop tests was grown on TSA at 22°C. For liquid cultures, *B. subtilis* was grown at 37°C in yeast extract-malt extract (YEME) medium overnight. *Streptomyces* strains were grown at 28°C in YEME medium for three days. Aliquots of the bacterial culture were pelleted, and the resulting supernatants were filtered through a 0.22 µm filter. Higher-density inocula required dilution. For example, after a 3 day incubation of *S. avermitilis*, the resulting cell-free supernatant was diluted 1:1000 with an initial inoculum of 13,000 spores/mL, whereas an inoculum of 650 spores/mL needed no dilution in the behavior assays. Activity in the cell-free supernatant degrades over time.

### Extraction and modification of fatty acids

A 20 mL aliquot of bacterial culture was pelleted, and the resulting supernatant was filtered through a 0.22 µm filter. The cell-free supernatant was extracted sequentially with a 1:5 mix of methanol/ methyl tert-butyl ether using a 40 mL then an 80 mL portion of the organic solvents. The combined organic layers were concentrated under reduced pressure. Fatty acid derivatization was performed according to Narayana (*Narayana et al., 2015*). Briefly, the extraction concentrate was dissolved in 100 µL acetonitrile (ACN). A 50 µL portion of a 1 mg/mL solution of carbonyldiimidazole in ACN was added, and the tube was inverted for mixing for 2 min. Next, 50 µL of a 50 mg/mL solution of 3-hydroxymethyl-1-methylpyridinium in 95:5 ACN/Et$_3$N was added to the reaction tube and inverted for mixing for an additional 2 min. The tube was heated at 40°C for 20 min. to complete the derivatization. A control of YEME medium spiked with decanoic acid was also extracted, derivatized and analyzed.

### UPLC/Electrospray Ionization-MS analysis

The LC-MS analysis was performed on an Agilent 6530 Accurate Mass QToF LC/MS with a dual electrospray ionization source (Agilent) connected to a Agilent 1290 Infinity II UPLC front end equipped with an SB C18 column (1.8 um, 3.0 × 100 mm, 828975–302 [Agilent]). The gas temperature was 350°C. The VCap was set to 4000V. The instrument was operated in positive ion mode under MS conditions. Absorbance threshold was set at 5000 and the relative threshold was set at 1%. Data were collected and stored in centroid form. The mobile phase system was made up of Solvent A (95:5 water:ACN + 0.1% formic acid) and Solvent B (95:5 ACN:water +0.1% formic acid), and a flow rate of 0.2 mL/min. The column was equilibrated for 10 min at 40% B before samples were run. The method and gradient used to elute the derivatized fatty acids were: 0' to 2' 40% B to 62% B, 2' to 3' held at 62% B, 3' to 5' 62% B to 100% B, 5' to 7' held at 100% B, 7' to 8' 100% B to 40% B, 8' to 13' 40% B. All gradients were linear. The first 1.2 min of the run were sent to waste to remove underivatized tags. MS data werecollected from 1.2 min until the end of the run. Analysis was performed using Agilent MassHunter Qualitative Analysis Software. Exact masses were calculated using ChemDraw and analyzing standards. Total ionization curves (TICs) were extracted using these exact masses ± 100 ppm, the preset default by MassHunter to generate electrospray ionization curves (EICs). EICs were manually integrated and the area under the peak was calculated by MassHunter.

### Phasmid (tail) avoidance assay

To examine the response of the phasmid neurons to repellents, tail dry-drop tests were performed as previously described (*Park et al., 2011*). Assays were performed on pre-dried plates (incubated overnight at 30°C), so that the drops quickly dry into the media, allowing sensation while eliminating wicking along the body of animals that might stimulate the head neurons. Briefly, a nose touch is administered to a day one gravid adult hermaphrodite to induce backward movement, then a capillary is used to administer a drop of either a control M13 buffer or an experimental sample behind

the worm. The time that the animal backs into the drop before stopping is recorded. For *Figures 1–2*, we tested the response of at least 40 adults in M13 buffer and at least 40 adults to 0.6 mM SDS (diluted in M13) on the same day. The relative response index was calculated by dividing the average backing time into the experimental treatment by the average backing time into 0.6 mM SDS, so as to normalize the SDS response index to 100%. For *Figure 4A and C*, the relative response index was calculated as previously described (*Park et al., 2011*). Briefly, for the mutant animals, we divide the average backing time into the experimental treatment by the average backing time into buffer, and normalized this ratio with the ratio of the average response time of wild-type animals to SDS and to buffer on the same day.

$$\textbf{wild type}: \frac{\frac{\text{Experimental treatment wild type}}{\text{Buffer wild type}}}{\frac{\text{SDS wild type}}{\text{Buffer wild type}}} = \frac{\text{Experimental treatment wild type}}{\text{SDS wild type}}$$

$$\textbf{Mutant}: \frac{\frac{\text{Experimental treatment mutant}}{\text{Buffer mutant}}}{\frac{\text{SDS wild type}}{\text{Buffer wild type}}}$$

For buffer bars in *Figure 4*, the experimental treatment was buffer.

Most of the replication in our studies, including the phasmid avoidance assays described in this section and the amphid avoidance assays and chemotaxis assays described below, was biological. Different worms are used for every measurement in a sample group, as animals might otherwise adapt to treatments or nose touches. While there may also be sources of technical variation, such as slight imprecision in time measurements, we observed little variation in backup time for individual worms moving into saline buffer, indicating that the amount of technical variation is lprobably very low.

## Amphid (head) avoidance assay

To examine the response of the amphid neurons to nonvolatile repellent, we adapted a behavioral assay developed by Hilliard and colleagues (*Hilliard et al., 2002*). As with the phasmid avoidance assay, tests were performed on pre-dried plates. A capillary was used to place a dry drop of either a control buffer or an experimental sample in front of a forward-moving worm. If a worm terminated forward movement and initiated backward movement after the head had contacted the dry drop area, we considered it a response. If the worm continued to move forward through the dry drop after the head region had moved into the area of the drop, then we considered that a failure to respond. Wild-type animals responded to the previously known repellent SDS the majority of the time. In the negative control buffer M13, the wild-type animals responded only approximately 20% of the time.

## Chemotaxis assay

To examine the response of nematodes to volatile cues, chemotaxis assays were conducted as previously described (*Bargmann et al., 1993*). Diacetyl was diluted in ethanol at 1:1000. For bacterial cultures, 1 µl of cultured bacteria was used as the experimental stimulant, whereas the medium in which the bacteria were grown was used as a control. Chemotaxis indexes were then calculated using the equation: $chemotaxis\ index = \frac{\#\,experimental - \#\,control}{\#\,total}$.

## Calcium imaging

Calcium-imaging experiments were conducted as previously described (*Krzyzanowski et al., 2013*). Briefly, microfluidic devices graciously provided by H. Gemrich and S. Kato at the University of California San Francisco were used to trap and immobilize the worms, exposing their nose to a stream of the control buffer (M13) or stimulant diluted in buffer (M13). Experimental animals were exposed first to buffer, then switched to stimulant after 15 s, then switched back to buffer after 60 s of exposure. Buffer control animals were instead exposed to buffer from the second channel (see Statistical Analysis). Fluorescent images were taken at 400X magnification every half second. ImageJ64 (*Abramoff et al., 2004*; *Rasband, 2009*) and the StackReg plugin were used to measure fluorescence intensity and to subtract background. The percent change in fluorescence intensity based on

the intensity at the 15 s timepoint was calculated. High-inoculum *S. avermitilis* extract (13,000 spores/mL) diluted 1:50 in M13 buffer and 1 mM dodecanoic acid in M13 were used.

## Statistical analysis

To calculate an appropriate sample size, we drew on our previous work using phasmid assays (*Park et al., 2011*), which showed that most effect sizes for mean back-up time comparisons are large. Using Cohen's d = 0.8 (which corresponds to a large effect), we calculated sample sizes of approximately 26 animals per group (for significance level alpha = 0.05, and power = 0.8). There was no reason to restrict the sample sizes for most genotypes, therefore we used samples sizes of at least 40 for most genotypes. From a pilot study using amphid assays, a representative effect size corresponded to sample proportions of approximately 0.3 and 0.8, leading to a sample size estimate of 14 animals per group (for significance level alpha = 0.05 and power = 0.8). Sample size estimates were obtained using the pwr package in R. There was no reason to restrict the sample sizes to 14 for most genotypes, therefore we used samples sizes of at least 40 for most genotypes.

Results are reported in the form of *P*-values in the figures (*, p<0.05; **, p<0.01; ***, p<0.001; NS, p>0.05), and exact *P*-values are reported in the source data files for the figures. *P*-values provide precise information about whether two populations differ significantly in either means or proportions.

For the amphid (head) avoidance assay, statistical significance was determined by two-independent-sample *z*-tests, followed by the Hochberg procedure. Independent sample z-tests can be used to compare two population proportions if the sample sizes are sufficiently large. The Hochberg procedure is a standard procedure applied to adjust for the tendency to reject a null hypothesis incorrectly when multiple comparison are made, and can only conservatively increase *P*-values. A multiple comparison procedure is required whenever several conclusions are drawn from the same group of data in order to assure that the overall error rate for a type I error is still bound by the chosen significance level (here 0.05, 0.01, 0.001). For phasmid (tail) response assays in *Figures 1* and *2, a* one-way ANOVA model to compare the means of three or more populations was fitted in R. Tukey's post-hoc test was performed in R to compare the means of the individual treatments while simultaneously adjusting for multiple comparisons. For the phasmid (tail) response assays reported in *Figure 4*, data representing mean length or means of percentages were analyzed with two-sample *t*-tests followed by the Hochberg procedure for multiple comparisons. For the pair-wise comparison of population means in the chemotaxis assays, Tukey's post-hoc test was used to adjust for multiple comparisons.

Effect sizes (Cohen's d) for major results include the following: wild-type phasmid response to *S. avermitilis* cell-free supernatant compared with buffer – 2.8; wild-type amphid response to *S. avermitilis* compared with buffer – 1.4; wild-type phasmid response to dodecanoic acid compared with buffer – 2.1; wild-type amphid response to dodecanoic acid compared with buffer –1.4; *srb-6* phasmid response to dodecanoic acid compared to wild-type phasmid response to dodecanoic acid – 1.8; *srb-6* amphid response to dodecanoic acid compared to wild-type amphid response to dodecanoic acid – 0.54.

For calcium-imaging experiments, we normalized the two different compound treatments to animals exposed to buffer in the first channel, then buffer in the second channel, then buffer in the first channel again (buffer control) by subtracting the averaged time series in the buffer control worms from each individual treatment time series. Then we computed the area under the curve during exposure to stimulus (15–75 s) for each individual normalized worm, and compared wild-type to *srb-6* using a two-sample t-test. We did this separately for the two different treatments.

## RNAi feeding experiments

The RNAi feeding screen was performed using the Ahringer library (*Fraser et al., 2000*) according to the Source BioScience protocol using the RNAi-sensitive strain *eri-1*(*mg366*) (*Kennedy et al., 2004*). L4s were picked onto plates seeded with each RNAi strain, and their adult progeny were tested for defects in responding to dodecanoic acid exposure at the tail. Assays were performed on at least two days for each RNAi clone.

## Cloning and transgenics

To generate the $_p$ocr-2::srb-6 construct, the srb-6 cDNA was amplified from an N2 cDNA library, adding 5' NcoI and 3' SacI sites, subcloned into the NcoI-SacI fragment of pSMdelta (a generous gift from S. McCarroll, Harvard University, Cambridge, MA), and sequenced to generate pSMdelta::srb-6. The srb-6 sequence was identical to that predicted on Wormbase.org (**WormBase web site, 2017**). The ocr-2 promoter was amplified from N2 genomic DNA, adding 5' SphI and 3' SmaI sites, and subcloned into the SphI-SmaI fragment of pSMdelta::srb-6 to generate $_p$ocr-2::srb-6. To generate transgenic strains, $_p$ocr-2::srb-6 and the $_p$odr-1::GFP (**37**) coinjection marker were injected into srb-6(gk925263) ox6 unc-4(e120) trans-heterozygotes, non-Unc GFP-positive $F_1$ hermaphrodites were singled out, $F_2$ plates with no Unc $F_2$ progeny were selected, and their progeny assayed for their response to S. avermitilis and dodecanoic acid and then sequenced. Defects in sensing S. avermitilis and dodecanoic acid were functionally assessed in non-array-carrying animals. To generate the calcium marker, the srb-6 promoter (**Troemel et al., 1995**) was amplified from genomic DNA, adding 5' AfeI and 3' XmaI sites, and inserted into the AfeI-XmaI fragment of flp-6::GCaMP6 to generate $_p$srb-6::GCaMP6. To generate transgenic strains, $_p$srb-6::GCaMP6 and the $_p$unc-122::DsRed coinjection marker were injected into unc-4(e120) heterozygotes. Wild-type and unc-4 mutants were isolated, srb-6(gk925263) ox6 was crossed into unc-4 mutants, srb-6 was homozygosed by pushing out the linked unc-4 gene, and the resulting lines were sequenced.

## Phylogenetic analysis

The phylogenetic tree in **Figure 1A** was constructed using the Maximum Likelihood method based on the Hasegawa-Kishino-Yano model (**Hasegawa et al., 1985**). The tree with the highest log likelihood is shown. The percentage of trees in which the associated taxa clustered together is shown next to the branches. Initial tree(s) for the heuristic search were obtained automatically by applying Neighbor-Join and BioNJ algorithms to a matrix of pairwise distances estimated using the Maximum Composite Likelihood (MCL) approach, and then selecting the topology with superior log likelihood value. The analysis involved five nucleotide sequences. There were a total of 1753 positions in the final dataset. Evolutionary analyses were conducted in MEGA7 (**Kumar et al., 2016**).

The C. elegans SRB-6 protein sequence record P54141.1 (aka NP_496199.1) was retrieved from NCBI. A pre-computed BLAST (RRID:SCR_004870) search was retrieved (https://www.ncbi.nlm.nih.gov/sutils/blink.cgi?pid=1711518). A BLAST search conducted against non-redundant database (nr) was re-run using the same settings. A dataset containing exactly 100 protein sequences was retrieved by BLAST with E-value <0.0001. A phylogenetic tree was built using the online web-server http://www.phylogeny.fr/

## Acknowledgements

We would especially like to thank M Fischbach for the use of equipment and facilities; H Gemrich and S Kato for microfluidic devices; J Engebrecht, L Rose and A Deshong for help with RNAi clones; J Ahringer for generation of RNAi strains; V Zavala for bacterial strain preparation; A Farooqi, J Griffin, F Farah, A Madrigal, S Pollock and M. Aguilar for assistance with experiments; S Bros, S Chalasani, E Glater, L Parr, S Rech, and E Skovran for comments and advice on the manuscript; the CGC and ARS Culture Collection for strains; Wormbase; and the National Institutes of Health (NIH) (NS087544 to MV and NL, GM089595 to MV and NS087544 to NL, 5SC3GM118199 to LMC, and 5T34GM008253 MARC fellowship to CW), the National Science Foundation (NSF) (1355202 to MV), and CSU-LSAMP (HRD-1302873 CSU-LSAMP fellowships to A. Tran and S. Matthews) for funding this work. CSU-LSAMP is funded through the National Science Foundation (NSF) under grant #HRD-1302873 and the Chancellor's Office of the California State University. Any opinions, findings, and conclusions or recommendations expressed in this material are those of the author(s) and do not necessarily reflect the views of the National Science Foundation or the Chancellor's Office of the CSU.

## Additional information

### Funding

| Funder | Grant reference number | Author |
| --- | --- | --- |
| National Science Foundation | HRD-1302873 | Angelina Tang<br>Sarah Y Matthews |
| NIH MARC Fellowship | 5T34GM008253 | Christopher Wellbrook |
| National Institutes of Health | NS087544 | Noelle L'Etoile<br>Miri K VanHoven |
| National Institutes of Health | 5SC3GM118199 | Laura C Miller Conrad |
| National Science Foundation | 1355202 | Miri K VanHoven |
| National Institutes of Health | GM089595 | Miri K VanHoven |

The funders had no role in study design, data collection and interpretation, or the decision to submit the work for publication.

### Author contributions

Alan Tran, Conceptualization, Formal analysis, Investigation, Visualization, Methodology, Writing—original draft, Writing—review and editing; Angelina Tang, Eric Chang, Doris Coto Villa, Sami Khuri, Conceptualization, Formal analysis, Investigation, Writing—original draft, Writing—review and editing; Colleen T O'Loughlin, Conceptualization, Formal analysis, Investigation, Methodology, Writing—original draft, Writing—review and editing; Anthony Balistreri, Joy Li, Aruna Varshney, Vanessa Jimenez, Jacqueline Pyle, Bryan Tsujimoto, Christopher Wellbrook, Christopher Vargas, Alex Duong, Nebat Ali, Samantha Levinson, Conceptualization, Investigation, Writing—original draft, Writing—review and editing; Sarah Y Matthews, Conceptualization, Investigation, Writing—review and editing; Sarah Woldemariam, Conceptualization, Methodology, Writing—review and editing; Martina Bremer, Conceptualization, Formal analysis, Methodology, Writing—original draft, Writing—review and editing; Daryl K Eggers, Conceptualization, Investigation, Methodology, Writing—original draft, Writing—review and editing; Noelle L'Etoile, Conceptualization, Funding acquisition, Methodology, Writing—original draft, Writing—review and editing; Laura C Miller Conrad, Conceptualization, Supervision, Investigation, Methodology, Writing—original draft, Writing—review and editing; Miri K VanHoven, Conceptualization, Supervision, Funding acquisition, Investigation, Visualization, Methodology, Writing—original draft, Project administration, Writing—review and editing

### Author ORCIDs

Miri K VanHoven ⓘD https://orcid.org/0000-0001-7714-1555

### Decision letter and Author response

Decision letter https://doi.org/10.7554/eLife.23770.019
Author response https://doi.org/10.7554/eLife.23770.020

## Additional files

### Supplementary files

• Transparent reporting form
DOI: https://doi.org/10.7554/eLife.23770.018

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
