## [Decision Letter]

Thank you for submitting your article "*C. elegans* avoids toxin-producing *Streptomyces* using a seven transmembrane domain chemosensory receptor" for consideration by *eLife*. Your article has been reviewed by two peer reviewers, and the evaluation has been overseen by Oliver Hobert as Reviewing Editor and a Senior Editor. The following individual involved in review of your submission has agreed to reveal her identity: Leslie B Vosshall (Reviewer #1).

The reviewers have discussed the reviews with one another and the Reviewing Editor has drafted this decision to help you prepare a revised submission.

The reviewers appreciate the description of a potential mechanism by which *C. elegans* nematodes detect and avoid toxic bacteria. The paper catalogues the species of bacteria that induce avoidance responses, use analytical chemistry to identify the bioactive repellents, and discover the sensory neurons and receptors responsible. The work is an engaging read on many levels: behavioral neuroscience, chemical ecology, and chemoreceptor biology.

The main experimental issue that needs to be addressed is based on the fact that the *srb-6* phenotype is based on a single allele and no rescue data is provided. Since the focus of *srb-6* action is also not explored, the reviewers suggest addressing both issues simultaneously by rescuing the *srb-6* phenotype with available neuron-type specific promoters. This would confirm that *srb-6* is indeed responsible for the phenotype and assess where the gene acts.

Other points that need to be fixed, but may not require further experimentation:

1) Statistical analysis throughout seems flawed. Rather than comparing everything to the saline buffer control and reporting significance relative to this, can the authors do ANOVA analysis and show us which of the data points differ from each other? Conventional presentation is to label plots that differ at some significance level with different letters.

2) It's not clear from the experiments what the actual link is between *Streptomyces* avoidance and dodecanoate (dodecanoic acid? see comment below) avoidance. First, most compounds that were tested elicited an avoidance response in this assay, and presumably many other bacterial metabolites that were not tested would also elicit a similar response. Second, what concentrations of metabolites were tested? This information does not appear to be provided. Was dose-response analysis performed?

3) There is confusion about the actual compounds that were tested. The authors refer to "dodecanoate" and "decanoate" as the tested compounds. But this isn't actually a compound. Does this mean sodium dodecanoate and sodium decanoate, or dodecanoic acid and decanoic acid? Based on the chemical structures in Figure 2'm guessing it was the acids that were tested. But then in the Discussion, "dodecanoate" is mentioned in the same paragraph as "dodecanoic acid" as if these are two different compounds. The authors need to specify exactly which chemicals they are testing and refer to them consistently throughout the paper. A table with CAS numbers would be helpful. Also, these compounds are referred to as "water-soluble cues" when I don't think they are very water-soluble.

4) Results paragraph three: This paragraph is confusing. Is there a reason to think that PHA and PHB only sense SDS and similar compounds? Also, how does the conclusion that *Streptomyces* is unlikely to act by physically disrupting the nematode follow from the fact that the surfactant activity of SDS is not required for avoidance? If PHA and PHB express multiple chemoreceptors, presumably they can sense a number of unrelated compounds.

5) Figure 2:

a) It would help to show molecular structures for C and E in addition to D.

b) It would be helpful to show statistics relative to the SDS positive control in addition to the buffer negative control. Based only on comparisons to the buffer control, it's hard to tell which compounds are as effective as SDS in eliciting an avoidance response.

c) In E, why are only a subset of the tested compounds listed in the legend? Also, why was a two-sample z-test used? I thought a z-test was for cases where you know there are two different populations embedded within the larger population (e.g. males vs. females).

6) Figure 3: The legend needs to be more detailed to be comprehensible to a non-chemist.

7) Figure 4: The authors mention that they did an expression screen, but no details are provided. Was this by transcriptional profiling? Using existing data available on WormBase? Were other receptors found that were also expressed in the PHA and PHB neurons in addition to a subset of amphid neurons? Since there is no information about the expression screen, it's not clear how unique *srb-6* really is. A rationale should be provided for why *srb-6* was picked for analysis.

8) Along the same lines as previous point: Have other GRCR mutants with similar expression pattern been tested for *Streptomyces* avoidance; if so, this data could be provided as control? Similarly, have other *srb* genes been tested? It would also be nice to see data for a screen of other *srb* mutants to see if the defect seen with *srb-6* is really specific to *srb-6*.

9) The authors show that *srb-6* mutants are defective in their response to S. avermitilis. Have *Streptomyces* species been tested? If *srb-6* is the primary receptor gene required for avoidance of *Streptomyces*, you'd expect *srb-6* mutants to be defective in their avoidance of all three species.

10) Figure 4—figure supplement 1: It would help to highlight *srb-6* so it is easier to find it in the phylogenetic tree. Also, aren't all of the srb genes nematode-specific?

11) Were the behavioral experiments performed across multiple days to control for day-to-day variation in behavior?

12) It would improve presentation clarity if the authors de-cluttered their plots by eliminated extraneous vertical lines to mark off the axis units

13) Figure 1—figure supplement 2 contains bar plots, which should be converted to box plots or dot plots so that readers can see something closer to the raw data

14) Figure 2 and Figure 3: hard to see the black text on top of the cyan semi-circle. Maybe play with color to enhance clarity

[Editors' note: further revisions were requested prior to acceptance, as described below.]

Thank you for submitting your revised article "*C. elegans* avoids toxin-producing *Streptomyces* using a seven transmembrane domain chemosensory receptor" for consideration by *eLife*. Your article has been reviewed by two peer reviewers and the evaluation has been overseen by a Reviewing Editor and a Senior Editor. The reviewers have discussed the reviews with one another and the Reviewing Editor has drafted this decision to help you prepare a revised submission.

It was generally agreed upon that the revisions did not properly address the concerns raised in the original review. The specific problems are as follows:

1) Figure 4 need to be modified to include data from the mutant. Mutant rescue cannot be interpreted without side-by-side data from the mutant vs. wild type vs. mutant rescue. From the graph, it looks like the authors did not test wt, *srb-6*, and *ocr-2::srb-6* in parallel. However, in the methods section, the authors say that they injected the rescue construct into *srb-6* mutants, and in the behavioral assays they tested "non-array carrying animals (data not shown)". Based on this statement, it's possible that the authors did test wt*, srb-6* (non-array carrying animals), and *ocr-2::srb-6* in parallel, but for some reason didn't show this. If this really is what they did, they should show the data and do the appropriate statistics to show that *srb-6* mutants are different from wt and the rescue line when tested in parallel. If this is not what they did, the authors need to repeat the experiment and test the three genotypes in parallel.

2) Figure 4: where is the missing "wild type buffer" control?

3) Since the authors have not individually rescued *srb-6* function in each of the 5 neurons in which it is expressed, they should soften language and interpretations about circuitry accordingly.

4) The authors' description of the rescue is ambiguous. In the main text, they say that they used a promoter that "drives expression selectively in PHA and PHB." However, they chose a promoter that is expressed in PHA, PHB, ASH, ADL, ADF, and AWA. The language implies to me that *srb-6* is only expressed in the phasmid neurons, but this is not the case. In the Materials and methods, they also talk about "cell-specific rescue," which again is ambiguous.

---

## [Author Response]

*The main experimental issue that needs to be addressed is based on the fact that the srb-6 phenotype is based on a single allele and no rescue data is provided. Since the focus of srb-6 action is also not explored, the reviewers suggest to address both issues simultaneously by rescuing the srb-6 phenotype with available neuron-type specific promoters. This would confirm that srb-6 is indeed responsible for the phenotype and assess where the gene acts.*

We agree that this is an interesting experiment. We cloned and sequenced the *srb-6* cDNA, which has a sequence identical to that predicted on Wormbase.org. Troemel et al., 1995 reported that *srb-6* is strongly expressed in PHA and PHB neurons in the tail, and ASH and ADL neurons in the head, and also weakly expressed in ADF neurons in the head, which are presynaptic to ASH. We therefore searched for a promoter that was expressed in ASH, ADL and ADF neurons and as few other pairs of head neurons as possible. The well characterized gene *ocr-2* is expressed in these three pairs of cells and only one other, the AWA olfactory neurons. *ocr-2* is additionally expressed in PHA and PHB neurons in the tail. We therefore generated an *pocr-2::srb-6* transgene, allowing us to perform cell-specific rescue assays in the head and tail with the same transgenic animals. The ability of *srb-6* mutants to respond to *S. avermitilis* and dodecanoic acid exposure at the tail and head, respectively, was rescued by addition of the *pocr-2::srb-6* transgene. These results are shown in Figure 4 and Figure 4 and are described in the Results section with the following text:

“In the tail, the limited expression of the *srb-6* GPCR in PHA and PHB neurons indicates that *srb-6* likely functions in these neurons to drive the tail response. […] Taken together, these data and previous results indicate that the circuit responsible for mediating the response to *Streptomyces* and dodecanoic acid exposure at the head may differ from the circuit responsible for the response to SDS.”

In the Discussion, we also added the following text:

“Rescue of gene expression in these cells and one additional amphid olfactory neuron largely restores the response of *srb-6* mutants to *Streptomyces* and dodecanoic acid exposure at the tail and head, respectively. […] While ASH, ADL and ASK mediate the response to SDS exposure at the head, ASH, ADL and ADF likely mediate the response to *Streptomyces* and dodecanoic acid exposure at the head, indicating distinct response circuits for these structurally similar compounds.”

In addition, *srb-6* RNAi produces a similar phenotype to that observed for the *srb-6* allele used in these studies (see response to reviewer comment 7), which should also help to address this concern.

*Other points that need to be fixed, but may not require further experimentation:*

*1) Statistical analysis throughout seems flawed. Rather than comparing everything to the saline buffer control and reporting significance relative to this, can the authors do ANOVA analysis and show us which of the datapoints differ from each other? Conventional presentation is to label plots that differ at some significance level with different letters.*

We performed 1-factor ANOVA analysis for relative response index data in Figure 1, Figure 2, and D, Figure 4, and Figure 4. In order to better illustrate the similarities and differences in backing times to SDS and buffer, respectively (as requested in the similar reviewer comment #5 part 2), we modified the appropriate figure panels to include both of these comparisons (Figure 1, Figure 2, Figure 2, and Figure 2). In addition, we now include tables of Tukey’s post-hoc test values for all pairwise comparisons for each of these figures in the source data files associated with these figures.

Since the worms utilized for the amphid assays are not the same in different treatment groups (e.g. buffer, SDS, dodecanoic acid, etc.), and we are interested in comparing proportions of worms that react to the treatment across the different groups, we are using independent sample Z-tests for proportions to establish significance. In order to better illustrate the similarities and differences of proportions to SDS and buffer, respectively (and address reviewer comment #5 part 2), we modified the appropriate figure panels to include both of these comparisons (Figure 1 and Figure 2). Additionally, we included tables of all pairwise comparisons for each of these figures in the source data files associated with these figures. All p-values are adjusted by the Hochberg method for multiple comparisons.

*2) It's not clear from the experiments what the actual link is between Streptomyces avoidance and dodecanoate (dodecanoic acid? see comment below) avoidance. First, most compounds that were tested elicited an avoidance response in this assay, and presumably many other bacterial metabolites that were not tested would also elicit a similar response.*

We propose in this manuscript that dodecanoic acid secreted by *Streptomyces* in a contributor to *C. elegans* avoidance of these toxin-producing species. It is likely that other secreted factors also contribute to this response. In support of this, we demonstrate that *C. elegans* avoids the species that secrete dodecanoic acid, species of *Streptomyces*, but do not avoid the control species that does not secrete dodecanoic acid, *B. subtilis*. We also identify a G protein-coupled receptor, *srb-6*, that is required for the response to both *Streptomyces* and dodecanoic acid. To clarify this for the readers, we have revised text in the Results: “While additional cues from *Streptomyces* likely contribute to the *C. elegans* response, our observation of dodecanoic acid supports the hypothesis that it is a cue produced by *Streptomyces*.” Also please note in the Discussion: “*C. elegans* rapidly avoids secretions from the bacteria, which include dodecanoic acid.”

To address the point that most compounds that were tested elicited an avoidance response in this assay, our goal was to understand the structure-activity relationship of compounds related to SDS, the most potent chemical activator of the phasmid neurons known, and a potent activator of ASH and ADL in the head in hermaphrodites. We predicted that most related compounds would have some activity. Still, we found that certain components were critical for detection. For example, when we removed the carboxylate or sulfate group (e.g. 1-dodecanol (Figure 2) or 1-decanol (Figure 2)), we lost all activity.

To address the point that presumably many other bacterial metabolites that were not tested would also elicit a similar response, we propose that the toxic *Streptomyces* secretion dodecanoic acid is avoided, but non-toxic secretions from bacteria that serve as a food source for *C. elegans* would be unlikely to elicit an escape response. In support of this, we show that *C. elegans* avoided secretions from *Streptomyces*, but they do not rapidly avoid secretions from *E. coli* or *B. subtilis.* However, as stated above, we agree that other secretions from *Streptomyces* likely contribute to the avoidance response. We have softened language in the paper to reflect the possibility that other bacterial secretions could be involved in this avoidance.

*Second, what concentrations of metabolites were tested? This information does not appear to be provided. Was dose-response analysis performed?*

Thank you for catching this accidental deletion. We have included concentration information to the captions of Figure 2 and Figure 4. Dose-response analysis for SDS, decanoic acid, and dodecanoic acid was also added as Figure 2—figure supplement 2.

*3) There is confusion about the actual compounds that were tested. The authors refer to "dodecanoate" and "decanoate" as the tested compounds. But this isn't actually a compound. Does this mean sodium dodecanoate and sodium decanoate, or dodecanoic acid and decanoic acid? Based on the chemical structures in Figure 2'm guessing it was the acids that were tested. But then in the Discussion, "dodecanoate" is mentioned in the same paragraph as "dodecanoic acid" as if these are two different compounds. The authors need to specify exactly which chemicals they are testing and refer to them consistently throughout the paper. A table with CAS numbers would be helpful. Also, these compounds are referred to as "water-soluble cues" when I don't think they are very water-soluble.*

We apologize for the confusion and thank the editors for their suggestion of adding CAS numbers. We have added a table with the CAS numbers (Figure 2—figure supplement 1). In addition, we used the terms for the carboxylic acids in the text, but have added the following statement to make it clear that at pH 7, the carboxylate form would be the predominant form in solution: “Both SDS and these carboxylic acids have long hydrophobic alkyl tails and negatively charged head groups at pH 7.” In the figures, we have maintained the terms and structures for the carboxylate forms, because these are the relevant forms when the compounds are dissolved in the pH 7 assay buffer. We feel that this is important to demonstrate the structural similarities among these molecules. For further clarity, in each appropriate figure panel (Figure 2, Figure 4) we added in the statement that compounds are represented as the predominant form in a pH 7 solution, the carboxylates. To prevent confusion, we also removed the three uses of the term “water-soluble” from the text.

*4) Results paragraph three: This paragraph is confusing. Is there a reason to think that PHA and PHB only sense SDS and similar compounds? Also, how does the conclusion that Streptomyces is unlikely to act by physically disrupting the nematode follow from the fact that the surfactant activity of SDS is not required for avoidance? If PHA and PHB express multiple chemoreceptors, presumably they can sense a number of unrelated compounds.*

We acknowledge that this point was not well explained in the original manuscript. We reworked this paragraph to clarify. It now reads:

“Previously, the PHA and PHB-class neurons in the tail were shown to act in a defined circuit respond to the man-made detergent SDS, which does not appear in nature. […] Therefore, we looked to define the structural elements of SDS that are responsible for the avoidance response.”

We do not think that PHA and PHB only sense SDS and similar compounds. However, SDS is the compound that elicits the most robust phasmid response known, and few other compounds have been fully characterized with respect to these neurons. This is why we chose it as a starting point for our studies. To make this more clear in the text, we added the following sentence: “Few compounds have been fully characterized with respect to these neurons, however dilute concentrations of sodium dodecyl sulfate (SDS) have been shown to elicit a rapid and robust avoidance response in hermaphrodites, the primary sex found in nature.”

Finally, we agree that PHA and PHB express multiple chemoreceptors, and presumably can sense a number of unrelated compounds. We explained why we focused on SDS and related compounds in this study above, but it would be very interesting to explore this large topic in future studies.

*5) Figure 2:*

*a) It would help to show molecular structures for C and E in addition to D.*

We have added in molecular structures for Figure 2 in addition to Figure 2.

*b) It would be helpful to show statistics relative to the SDS positive control in addition to the buffer negative control. Based only on comparisons to the buffer control, it's hard to tell which compounds are as effective as SDS in eliciting an avoidance response.*

Please see the response to reviewer comment #1, which addresses this directly. In summary, we have added in statistics relative to the SDS positive control in addition to the buffer negative control for all appropriate Figure panels.

*c) In E, why are only a subset of the tested compounds listed in the legend? Also, why was a two-sample z-test used? I thought a z-test was for cases where you know there are two different populations embedded within the larger population (e.g. males vs. females).*

We have now listed all compounds in the legend for Figure 2. For the question regarding the use of the two-sample z-test, please see the second paragraph in the response to reviewer comment #1, which addresses this directly.

*6) Figure 3: The legend needs to be more detailed to be comprehensible to a non-chemist.*

More detail has been added to the Figure 3 legend, and for clarity, we have also added labels into the figure. The legend now reads:

“Figure 3.Detection of carboxylates secreted by *S. avermitilis, S. costaricanus,* and *S. milbemycinicus.* Carboxylates were extracted from species of *Streptomyces* and labeled with a pyridinium moiety for easier detection by mass spectrometry. For example, to detect dodecanoate, after *Streptomyces* cell-free supernatants were extracted and concentrated, dodecanoate and the other carboxylates in the extract were converted to pyridinium esters by treatment with carbonyldiimidazole (CDI) in acetonitrile (ACN).”

*7) Figure 4: The authors mention that they did an expression screen, but no details are provided. Was this by transcriptional profiling? Using existing data available on WormBase? Were other receptors found that were also expressed in the PHA and PHB neurons in addition to a subset of amphid neurons? Since there is no information about the expression screen, it's not clear how unique srb-6 really is. A rationale should be provided for why srb-6 was picked for analysis.*

Using existing data available on WormBase, we screened for receptors expressed in the PHA and PHB neurons in addition to a subset of amphid neurons. The majority of these genes do not have characterized loss-of-function alleles, therefore we used RNAi to screen for behavioral defects. *srb-6* RNAi had the most severe defect. We have added Figure 4—figure supplement 1 to present the results of this RNAi screen. We also revised the appropriate section of the Results to describe the screen. The text now reads:

“To identify the cognate receptor, we performed an expression screen using existing data available on Wormbase.org for chemoreceptors that are expressed in the PHA and PHB neurons and at least one set of amphid neurons. […] We ranked them by the magnitude of the defect, and focused our work on the gene whose knock-down resulted in the most severe phenotype, namely the *srb-6* G protein-coupled receptor (GPCR) (Figure 4—figure supplement 1).”

*8) Along the same lines as previous point: Have other GRCR mutants with similar expression pattern been tested for Streptomyces avoidance; if so, this data could be provided as control? Similarly, have other srb genes been tested? It would also be nice to see data for a screen of other srb mutants to see if the defect seen with srb-6 is really specific to srb-6.*

For testing of other GPCRs with similar expression patterns please see our response to reviewer comment #7. To address the question about other *srb* genes, only one *srb* gene has previously reported phasmid expression, *srb-3*, but RNAi of *srb-3* did not yield a severe behavioral defect in our RNAi screen (see response to reviewer comment #7). However to further address this comment, in case expression in the phasmid was not reported, we obtained all RNAi strains available for *srb* genes from the Ahringer collection (excluding pseudogenes) and performed RNAi. *srb-6* again had the most severe defect. The data has been included in Figure 4—figure supplement 1. We added the following sentence to the Results section:

“*srb-6* similarly had the most severe defect within the *srb* family of genes (Figure 4—figure supplement 1).”

*9) The authors show that srb-6 mutants are defective in their response to S. avermitilis. Have Streptomyces species been tested? If srb-6 is the primary receptor gene required for avoidance of Streptomyces, you'd expect srb-6 mutants to be defective in their avoidance of all three species.*

We tested *srb-6* mutants for their response to exposure to *S. costaricanus,* and *S.milbemycinicus* at the tail and head and found *srb-6* mutants to be defective in their avoidance of these species. We have added this data into Figure 4 and Figure 4.

*10) Figure 4—figure supplement 1: It would help to highlight srb-6 so it is easier to find it in the phylogenetic tree. Also, aren't all of the srb genes nematode-specific?*

We have highlighted *srb-6*, its closest homologs within the *Caenhorabditis* genus, and the monophyletic group containing nematodes outside the *Caenhorabditis* genus in different colors (now Figure 4—figure supplement 2, entitled “*C. elegans srb-6* has homologs within the *Caenhorabditis* genus and the nematode phylum”). We have also revised and moved the section describing the tree to attempt to make our purpose more clear. The section is now at the end of the Results and reads:

“These circuits in the head and tail of *C. elegans* and the GPCR required for the avoidance response provide *C. elegans* with the machinery needed to avoid potentially deadly prey. […] Although the sequences for most nematodes are not complete, the next monophyletic group contains several other species of nematodes with life cycles that include free-living soil-dwelling stages (Figure 4—figure supplement 2), suggesting possible conservation of this mechanism.”

*11) Were the behavioral experiments performed across multiple days to control for day-to-day variation in behavior?*

The sequence of behavioral experiments was performed across several years. In the phasmid assay, while absolute back-up times into both buffer and compounds are variable from day to day, the ratio of back-up times between SDS and buffer, between other compounds and buffer, and between different compounds and SDS, stays fairly consistent. Therefore to control for day-to-day variation, we work with ratios of individual back-up times in a given compound compared to average backup times in SDS. In the amphid assay, not much variation from day to day was observed for proportions of animals that responded to SDS or buffer. However, both RNAi and chemotaxis assays performed in Figure 1—figure supplement 2 and Figure 4—figure supplement 1 are variable from day-to-day due to variable success of RNAi and possible other sources of variation in chemotaxis assays, so these experiments were performed on more than one day in order to be able to quantify day-to-day variation.

*12) It would improve presentation clarity if the authors de-cluttered their plots by eliminated extraneous vertical lines to mark off the axis units*

We have removed vertical lines to mark off the axis units from all plots.

*13) Figure 1—figure supplement 2 contains bar plots, which should be converted to box plots or dot plots so that readers can see something closer to the raw data*

The figure has been converted into a dot plot.

*14) Figure 2 and Figure 3: hard to see the black text on top of the cyan semi-circle. Maybe play with color to enhance clarity*

We have lightened the cyan color, and are happy to lighten it further if the text is still hard to see on your screen.

We have one additional note. When we performed the cell-specific rescue experiments requested by the reviewers, we used new *Streptomyces* preparations, and the response rate in the head was a little higher in wild-type animals, possibly due to improved growth conditions. We have revised the data in Figure 1 to be consistent with the new data in Figure 4, but no text changes were required, as this does not change the interpretation.

[Editors' note: further revisions were requested prior to acceptance, as described below.]

*1) Figure 4 need to be modified to include data from the mutant. Mutant rescue cannot be interpreted without side-by-side data from the mutant vs. wild type vs. mutant rescue. From the graph, it looks like the authors did not test wt, srb-6, and ocr-2::srb-6 in parallel. However, in the Materials and methods section, the authors say that they injected the rescue construct into srb-6 mutants, and in the behavioral assays they tested "non-array carrying animals (data not shown)". Based on this statement, it's possible that the authors did test wt, srb-6 (non-array carrying animals), and ocr-2::srb-6 in parallel, but for some reason didn't show this. If this really is what they did, they should show the data and do the appropriate statistics to show that srb-6 mutants are different from wt and the rescue line when tested in parallel. If this is not what they did, the authors need to repeat the experiment and test the three genotypes in parallel.*

We have tested wild type, *srb-6* (non-transgene) and *ocr-2::srb-6; srb-6* in parallel, and performed the appropriate statistics to show that *srb-6* mutants are different from wild-type and the rescue lines when tested in parallel, see Figure 4 (previously Figure 4) and figure legend.

*2) Figure 4: where is the missing "wild type buffer" control?*

We thank the reviewer for pointing out this potential point of confusion. We have added a wild type buffer control to Figure 4 (previously 4D) and we have revised our Materials and methods section to make more clear that the relative response index measurements, which we previously simply cited, include a normalization to buffer controls (Statistical analysis, final paragraph).

*3) Since the authors have not individually rescued srb-6 function in each of the 5 neurons in which it is expressed, they should soften language and interpretations about circuitry accordingly.*

We agree and have revised the text accordingly. Specifically, we have softened the language and interpretations about circuitry throughout the manuscript. Major examples include the following: in the Results, we are now careful to say that one or a subset of the cells mediates the response in the tail and in the head. Similarly, in the Discussion, instead of discussing a model in which all of the *srb-6* expressing neurons function in the same circuit, we instead mention chemosensory roles of each sensory neuron in which *srb-6* is expressed. These changes streamlined the focus of the paper and we thank you for suggesting them. In addition, we performed calcium imaging experiments on ASH neurons in wild-type and *srb-6* mutant animals, and found that *srb-6* mutants have significantly reduced calcium influx in response to both *S. avermitilis* and dodecanoic acid, thus specifically implicating this neuron in the head and indicating that *srb-6* can act as a receptor for both stimuli (Figure 4).

*4) The authors' description of the rescue is ambiguous. In the main text, they say that they used a promoter that "drives expression selectively in PHA and PHB." However, they chose a promoter that is expressed in PHA, PHB, ASH, ADL, ADF, and AWA. The language implies to me that srb-6 is only expressed in the phasmid neurons, but this is not the case. In the Materials and methods, they also talk about "cell-specific rescue," which again is ambiguous.*

We thank the reviewers for drawing this potential point of confusion to our attention. We were trying to convey that within the tail region, the promoter selectively drives expression in PHA and PHB neurons. Our previous phrasing, “[…] we cloned the *srb-6* cDNA and expressed it under the direction of a promoter that drives expression selectively in the PHA and PHB neurons in the tail region (p*ocr-2*)” could be ambiguous. Therefore we have clarified to the following statement, “[…] we cloned the *srb-6* cDNA and expressed it under the direction of the *ocr-2* promoter. Within the tail region, this promoter selectively drives expression in the PHA and PHB neurons (40).” It is important to note that in the next paragraph we describe the use of the *ocr-2* promoter to rescue *srb-6* function in chemosensory neurons in the head, and also use the name of the *ocr-2* promoter in Figure 4 and in the Materials and methods. Just in case there might be any similar confusion about expression of the *ocr-2* promoter in the head region, we also moved “Within the head” to the beginning of the sentence/paragraph to clarify; the text now reads, “In the head, the ocr-2 promoter drives expression in ASH, ADL and ADF gustatory neurons in addition to AWA olfactory neurons.” Also, in the Materials and methods, we removed the word “cell-specific”. The sentence now reads, “Transgenes used for rescue of *srb-6* and maintained as extrachromosomal arrays are iyEx348 and iyEx350*”*